# Global Phylogeny and F Virulence Plasmid Carriage in Pandemic *Escherichia coli* ST1193

Ethan R. Wyrsch,[a] Rhys N. Bushell,[b] Marc S. Marenda,[b] Glenn F. Browning,[b] Steven P. Djordjevic[a]

[a]Australian Institute for Microbiology & Infection, University of Technology Sydney, Ultimo, New South Wales, Australia
[b]Asia-Pacific Centre for Animal Health, Department of Veterinary Biosciences, Melbourne Veterinary School, Faculty of Veterinary and Agricultural Sciences, University of Melbourne, Parkville, Victoria, Australia

**ABSTRACT** Lower urinary tract, renal, and bloodstream infections caused by phylogroup B2 extraintestinal pathogenic *Escherichia coli* (ExPEC) are a leading cause of morbidity and mortality. ST1193 is a phylogroup B2, multidrug-resistant sequence type that has risen to prominence globally, but a comprehensive analysis of the F virulence plasmids it carries is lacking. We performed a phylogenomic analysis of ST1193 ($n$ = 707) whole-genome sequences from EnteroBase using entries with comprehensive isolation metadata. The data set comprised isolates from humans ($n$ = 634 [90%]), including 339 (48%) from extraintestinal infection sites, and isolates from companion animals, wastewater, and wildlife. Phylogenetic analyses combined with gene detection and genotyping resolved an ST1193 clade structure segregated by serotype and F plasmid carriage. Most F plasmids fell into one of three related plasmid subtypes: F⁻: A1:B10 ($n$ = 444 [65.97%]), F⁻:A1:B1 ($n$ = 84 [12.48%]), and F⁻:A1:B20 ($n$ = 80 [11.89%]), all of which carry the virulence genes *cjrABC* colocalized with *senB* (*cjrABC-senB*), a trademark signature of F29:A⁻:B10 subtype plasmids (pUTI89). To examine the phylogenetic relationship of these plasmids with pUTI89, complete sequences of F⁻:A1:B1 and F⁻:1:B20 plasmids were resolved. Unlike pUTI89, the most dominant and widely disseminated F plasmid that carries *cjrABC-senB*, F plasmids in ST1193 often carry a complex resistance region with an integron truncation ($intI1_{\Delta745}$) signature embedded within a structure assembled by IS*26*. Plasmid analysis shows that ST1193 has F plasmids that carry *cjrABC-senB* and ARG-encoding genes but lack *tra* regions and are likely derivatives of pUTI89. Further epidemiological investigation of ST1193 should seek to confirm its presence in human-associated environments and identify any potential agricultural links, which are currently lacking.

**IMPORTANCE** We have generated an updated ST1193 phylogeny using publicly available sequences, reinforcing previous assertions that *Escherichia coli* ST1193 is a human-associated lineage, with many examples sourced from human extraintestinal infections. ST1193 from urban-adapted birds, wastewater, and companion animals are frequent, but isolates from animal agriculture are notably absent. Phylogenomic analysis identified several clades segregated by serogroup, all noted to carry highly similar F plasmids and antimicrobial resistance (AMR) signatures. Investigation of these plasmids revealed virulence regions with similarity to pUTI89, a key F virulence plasmid among dominant pandemic extraintestinal pathogenic *E. coli* lineages, and encoding a complex antibiotic resistance structure mobilized by IS*26*. This work has uncovered a series of F virulence plasmids in ST1193 and shows that the lineage mimics the host range and virulence attributes of other *E. coli* strains that carry pUTI89. These observations have significant ramifications for epidemiological source tracking of emerging and established pandemic ExPEC lineages.

**KEYWORDS** antimicrobial resistance, *Escherichia coli*, integron, pandemic, plasmids, virulence

Address correspondence to Steven P. Djordjevic, Steven.Djordjevic@uts.edu.au.

The authors declare no conflict of interest.

10.1128/spectrum.02554-22   1

Sequence type 1193 (ST1193) is a fluoroquinolone-resistant lineage of *Escherichia coli* phylogroup B2, within the ST14 clonal cluster (1, 2). Its closest relative is *E. coli* ST6460 (3). ST1193 is considered a hybrid lineage and is estimated to have arisen in 2005 by acquisition, via conjugation, of a single 976-kb chromosomal fragment from a distantly related ST10 *fimH*54 strain of *E. coli*. The conjugation event is remarkable because subsequent recombination events resulted in acquisition of multiple mutations in *gyrA* and *parC* and consequent resistance to fluoroquinolones (3–5). ST1193 isolates' shared features include three chromosomal mutations conferring fluoroquinolone resistance, a *fimH*64 allele, and a non-lactose-fermenting phenotype (2).

ST1193 is known to carry diverse plasmid cargo, including large multidrug resistance plasmids (6), and the first carbapenem nonsusceptible ST1193 isolates carrying $bla_{NDM}$ (7) and $bla_{KPC}$ (8) were reported recently. The sequence type is remarkable because it has risen to prominence in a short period of time and has been recovered from multiple hosts, including humans (9, 10), dogs (11, 12), synanthropic birds (13, 14), and water sources (6). While it is likely that humans are an important gastrointestinal reservoir for ST1193, dogs are considered significant in the human-companion animal infection chain (1).

Reports of infections with ST1193 surfaced in 2012 from health care settings in Germany (15), England (16), France (17), Canada (18), China (10, 19, 20), the United States (21), South Korea (9), and Australia (11, 12, 22), raising awareness of its importance as a rapidly emerging human pathogen, but few studies have examined the lineage from a One Health perspective. In humans, ST1193 strains are a cause of urinary tract (3), bloodstream (9, 18, 21), and other extraintestinal and invasive infections, and many are multidrug resistant. In China, ST1193 is a cause of invasive infections in neonates, with most of the invasive isolates being resistant to ciprofloxacin, amoxicillin, sulfonamides, and tetracycline (10). Notably, a third of the invasive ST1193 isolates studied had acquired $bla_{CTX-M-15}$, which encodes an extended spectrum of resistance to $\beta$-lactams (10). ST1193 can also persistently colonize the gastrointestinal tract of healthy women (3) but was not identified in the feces of healthy women in France prior to 2010 (23).

Australia was one of the first places to report ST1193 as a cause of extraintestinal disease in humans and dogs (11). Apart from being fluoroquinolone resistant, the Australian ST1193 isolates belonged to serogroup O75 and clonal cluster 14, displaying resistance to multiple antimicrobials, including piperacillin, gentamicin, tetracycline, and trimethoprim-sulfamethoxazole (11). The presence of these antimicrobial resistance genes (ARGs) indicates that ST1193 likely hosts class 1 integron structures located within complex resistance regions that are assembled by insertion elements, particularly IS*26* (24), which play important roles in the mobilization and capture of ARGs within *E. coli*.

One of the key influencers of extraintestinal pathogenicity in *E. coli* is F plasmid carriage, an assertion supported by the repeated observations of key virulence factors, particularly siderophores, which assist in iron capture during urinary tract infection (UTI) (25, 26), being mobilized by F family plasmids (27–30). An F plasmid replicon sequence typing (RST) scheme (31) allows subtyping of F plasmids from whole-genome sequence data, primarily with typing of RNAI-FII (designated "F"), *repFIA* ("A"), and *repFIB* ("B") alleles, reported as F:A:B. Two particular groups of F virulence plasmid are widely associated with pandemic extraintestinal pathogenic *E. coli* (ExPEC) lineages: F29:A⁻:B10 plasmids, typified by pUTI89 and carrying the predicted iron uptake system genes *cjrABC-senB* (26), and the comparatively broader family of ColV plasmids, characterized by the presence of numerous virulence operons (colicin [*cva*], salmochelin [*iro*], aerobactin [*iuc*], *ets*, *sit*, *ompT*, and *hlyF*) (32), again primarily iron uptake systems, and typed frequently as F2, F18, and F24 plasmids (33). F29:A⁻:B10 is one of the most frequently observed F RSTs and is widespread in ExPEC isolates from humans, wastewater, and companion and wild animals but is conspicuously missing in ExPEC from animal agriculture (28). ColV plasmids are also widespread in ExPEC isolates from diverse sources and have a notable link with animal agriculture, particularly poultry

and swine. Several chromosomal islands are also significant contributors to pathogenicity in UTI models (34), particularly the high-pathogenicity island (HPI) carrying the yersiniabactin siderophore genes (*irp* and *fyuA*) (35) and the colibactin operon (36).

The perception of public health is being impacted by One Health approaches to infectious disease (37). Here, we undertook a comprehensive phylogenetic analysis of 707 *E. coli* ST1193 isolates from different countries and multiple sources, contributing two completed sequences from Australian companion animals and providing insight into the role of F plasmids in the success of *E. coli* ST1193 as a pandemic pathogen. Collectively, our analysis has consequences for how emerging *Escherichia coli* pathogens are identified and traced.

## RESULTS

The genome sequences of ST1193 isolates (*n* = 707) depicted here were retrieved from EnteroBase and were selected based on having available isolation source, collection year, and country of isolation metadata. A further 553 genomes were excluded because metadata were incomplete. Antimicrobial resistance (AMR) and virulence genotypes, F plasmid RST subtypes, and carriage of the class 1 integrase gene *intI1* and insertion sequence IS*26* were also included. The *intI1* allele used for this analysis can be found under accession number MZ396394.1. Using a different *intI1* sequence can alter the reported hit sizes for this analysis by several base pairs, depending on the insertion site of IS*26*. The complete data set is available in Table S1 in the supplemental material. Geographically, the isolates were from 32 countries, with Australia (*n* = 259 [36.5%]) and the United States (*n* = 135 [19.0%]) being the most prominent contributors of sequence data. Isolates were primarily from humans (*n* = 634 [89.7%]), frequently with a clinical presentation. The remainder were from companion animals (canine and feline), wild avian sources, wastewater, wastewater treatment plants (WWTP), and other water sources. As of 31 August 2022, EnteroBase included approximately 44,000 entries reported from livestock, animal feed, and food products, and despite this wealth of data, ST1193 genomes from agricultural sources were notably absent outside a single entry from bovine livestock. This isolate was excluded from analysis due to a lack of metadata (Table S2). The earliest available ST1193 sequence was from an Australian isolate obtained in 2007. Table 1 presents key metadata and genetic features of ST1193 isolates from countries that contributed at least 10 isolates.

A phylogenetic tree based on single nucleotide variations (SNVs) was generated and aligned with isolate metadata and genotyping (Fig. 1). The clade structure segregated according to serotype (where serotype could be identified), with clusters comprising types O75:H5 (*n* = 676), O45:H5 (*n* = 5), and O25:H5 (*n* = 3), clade O18:H5 (*n* = 13), and clade O6:H5 (*n* = 10) identified. ST1193 isolates in serogroup O75:H5 were the dominant lineage. Sublineages within this serogroup fell into three groups associated with F plasmid RST. The majority of O75:H5 clade isolates, as well as O6:H5 and O45:H5 clade isolates, were predicted to carry F⁻:A1:B10 plasmids. F⁻:A1:B1 plasmids were identified in O18:H5 and O25:H5 isolates, as well as a subclade of O75:H5, and sporadically in other clades. F⁻:A1:B20 plasmid carriage was generally limited to a discrete subclade of the O75:H5 isolates. In all cases, these differently typed subclades comprised isolates from multiple nations and sources, indicating that at least seven (eight if the distant O75:H5 group is included) discernible subgroups of ST1193 are in circulation globally.

While 673 of 707 isolates (95%) carried an F plasmid sequence represented by 34 plasmid subtypes, 29 of these subtypes were detected in five or fewer isolates. F⁻:A1:B10 plasmids were identified in 444/673 (65.97%) isolates, followed by F⁻:A1:B1 (*n* = 84; 12.48%) and F⁻:A1:B20 (*n* = 80; 11.89%) plasmids, and these three plasmid replicon types represented 90.34% of all isolates carrying an F plasmid. Many of the remaining plasmid subtypes appear to be related to this set of three plasmid RSTs, often having a novel FII RST subtype or lacking the FIA subtype (e.g., F95:A1:B20 or F⁻:A⁻:B10). Notably, 591 isolates carried the virulence factor gene *senB*, with BLASTn data

**TABLE 1** Key features of *Escherichia coli* ST1193

| Country (n)[a] | Source(s) (n)[b] | e-serotypes (n) | Isolation date(s) | intI1 hits (bp) (n)[c] | F plasmid RSTs (n)[c] |
|---|---|---|---|---|---|
| Australia (259) | Human (237), wild avian (12), canine (10) | O75:H5 (118), O18:H5 (7), O6:H5 (5), O-:H5 (129) | 2007–2020 | 1,014 (57), Δ746 (108), Δ850 (12), Δ798 (2), Δ682 (2), Δ548 (2), other (3) | F-:A1:B10 (170), F-:A1:B20 (46), F-:A1:B1 (24), F-:A-:B10 (3), F-:A1:B- (2), other (8) |
| Canada (41) | Human (38), canine (1), water (2) | O75:H5 (34), O-:H5 (7) | 2010–2019 | 1,014 (12), Δ746 (13), other (1) | F-:A1:B10 (27), F-:A1:B1 (6), other (4) |
| China (46) | Human (46) | O75:H5 (40), O18:H5 (1), O25:H5 (3), O-:H5 (2) | 2011–2018 | Δ746 (21), other (3) | F-:A1:B10 (20), F-:A1:B1 (12), F-:A1:B20 (2), F12:A1:B10 (2), other (4) |
| Denmark (13) | Human (13) | O75:H5 (9), O-:H5 (4) | 2018 | 1,014 (5), Δ746 (3) | F-:A1:B10 (7), F-:A1:B20 (2), other (4) |
| France (12) | Human (10), canine (1), feline (1) | O75:H5 (11), O-:H5 (1) | 2014–2020 | Δ746 (9), other (1) | F-:A1:B10 (9), F-:A1:B20 (2), other (1) |
| Japan (11) | Human (4), water (7) | O75:H5 (9), O-:H5 (2) | 2015–2018 | 1,014 (1), Δ746 (4) | F-:A1:B10 (5), F-:A1:B20 (2), F-:A1:B1 (2), other (1) |
| New Zealand (23) | Human (23) | O75:H5 (16), O-:H5 (7) | 2015–2016 | 1,014 (1), Δ746 (8), other (1) | F-:A1:B10 (20), other (3) |
| Qatar (11) | Human (11) | O75:H5 (10), O-:H5 (1) | 2016–2018 | 1,014 (1), Δ850 (4), other (1) | F-:A1:B10 (8), F-:A1:B- (2), other (1) |
| Singapore (36) | Human (34), ND (2) | O75:H5 (24), O6:H5 (2), O-:H5 (10) | 2012–2018 | 1,014 (7), Δ746 (11), Δ850 (2), Δ682 (2), other (1) | F-:A1:B10 (28), F-:A1:B1 (3), other (4) |
| Switzerland (14) | Human (11), water (2), ND (1) | O75:H5 (9), O6:H5 (1), O-:H5 (4) | 2014–2020 | 1,014 (1), Δ746 (5), other (3) | F-:A1:B20 (7), F-:A1:B10 (5), other (2) |
| Thailand (16) | Human (16) | O75:H5 (11), O45:H5 (4), O18:H5 (1) | 2014–2018 | 1,014 (2), Δ746 (10), other (1) | F-:A1:B10 (12), other (2) |
| United Kingdom (15) | Human (15) | O75:H5 (13), O18:H5 (1), O-:H5 (1) | 2010–2021 | 1,014 (4), Δ746 (2), other (2) | F-:A1:B10 (8), F-:A1:B20 (2), F35:A1:B10 (2), other (2) |
| United States (135) | Human (125), canine (7), ND (3) | O75:H5 (66), O-:H5 (69) | 2009–2021 | 1,014 (34), Δ746 (20), Δ667 (3), Δ666 (15), other (7) | F-:A1:B10 (73), F-:A1:B1 (28), F-:A-:B10 (10), F-:A1:B20 (7), other (8) |
| Vietnam (34) | Human (12), ND (22) | O75:H5 (19), O-:H5 (15) | 2014–2018 | 1,014 (1), Δ746 (28), other (1) | F-:A1:B10 (26), F-:A1:B20 (3), F-:A1:B1 (2), other (3) |

[a]Only counts of 10 or greater are included.
[b]ND, not described.
[c]Those with counts of 1 are grouped as "other," with 1,014-bp *intI1* being an exception. Note that the 746-bp *intI1* hit reported has a corrected annotation size of 745 bp.

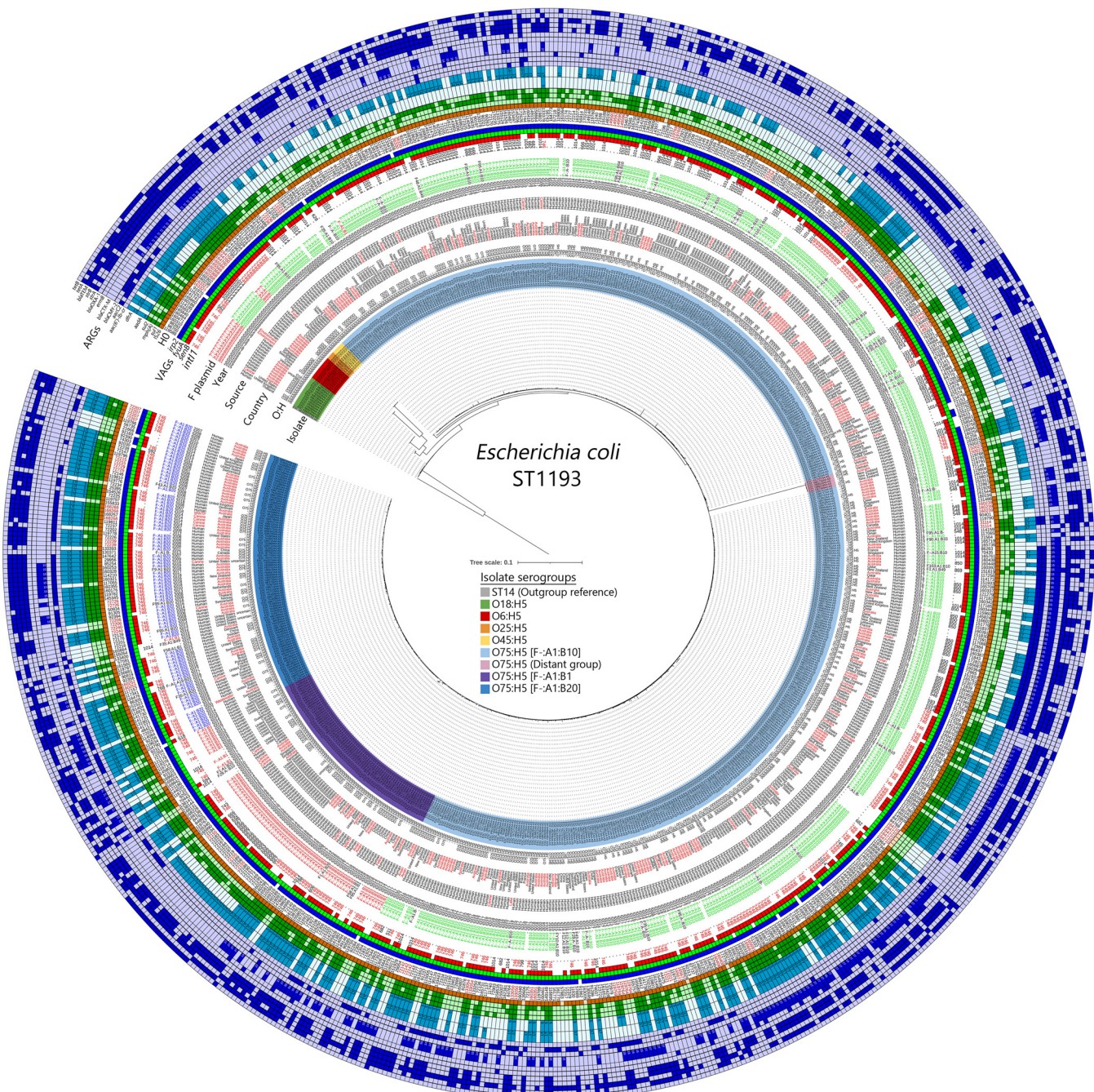

**FIG 1** Phylogenetic core SNV maximum-likelihood tree of *E. coli* ST1193 genome sequences. The tree is rooted on a representative of ST14, *E. coli* assembly ESC_NB5808AA_AS. Data placed at isolate tips include (in order) isolate metadata (country, source, and year) and genotypic information (serogroup [O:H], F plasmid subtype, *intl1* BLASTn hit size, and presence of *senB*, *fyuA*, and *irp2*), HierCC data at H0 (related isolates at H0 are highlighted), and ARG presence. Highlighted data included the most common country source (Australia), sources other than human, the most common *intl1* BLASTn hit, and F plasmid subtypes in their respective rings. The tree scale is in substitutions per site. A group of O75:H5 isolates that were notably distant from the rest of the serogroup is highlighted in pink.

confirming colocalization with *cjrABC* (*cjrABC-senB*), as was originally described and characterized in pUTI89 (F29:A⁻:B10). Within the whole collection, only one isolate encoding *senB* lacked an F plasmid, consistent with observations of *senB* being primarily distributed on F plasmids together with *cjrABC*. Interestingly, just two isolates were predicted to carry an F29:A⁻:B10 plasmid.

Siderophore genes detected included *fyuA* ($n = 703$) and *irp2* ($n = 690$), which are associated with the *Yersinia* HPI, and the *sit* operon ($n = 705$), which is associated with

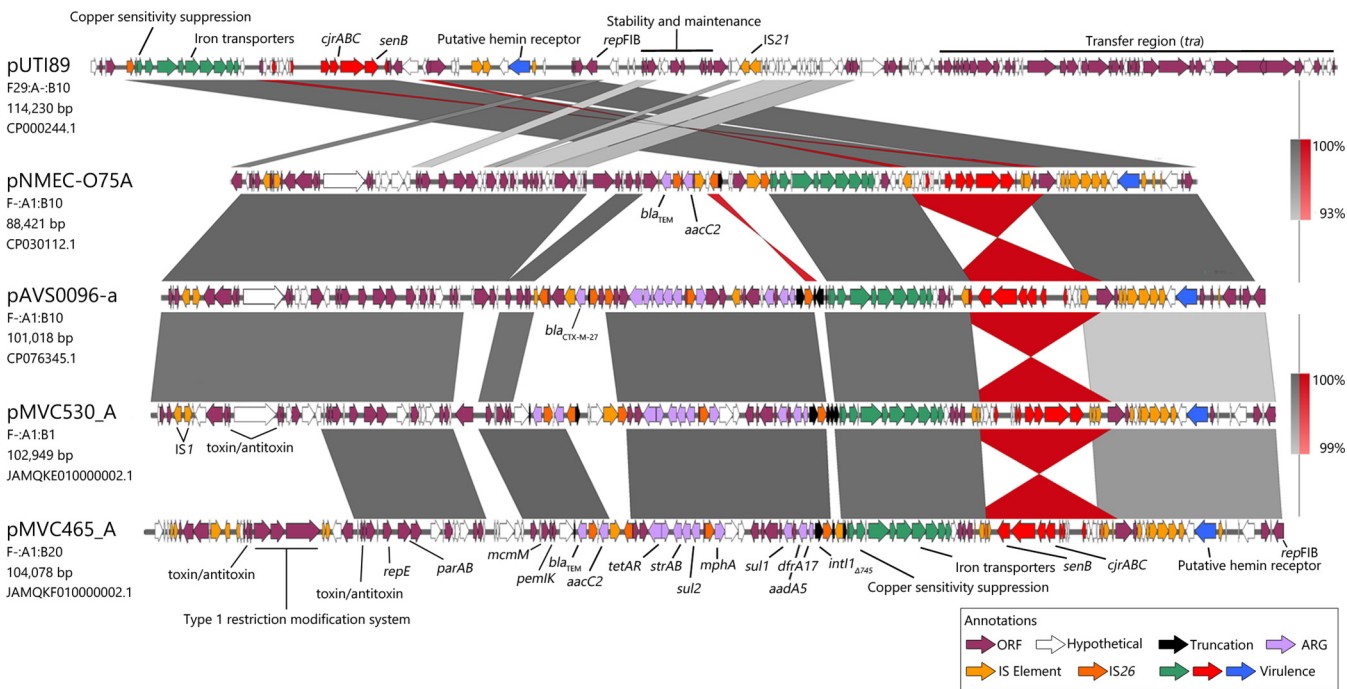

**FIG 2** Comparative linear alignments of plasmids from ST1193 with pUTI89. BLASTn identity is split between the two top and bottom sets of alignments, with the range indicated to the right of each pair. Note that some ORF annotations are hidden due to overlap, particularly for IS elements. Hits were restricted to 1,000 bp or more.

the *iut-iuc* (aerobactin) operon (*n* = 694). The *iro* operon was detected in only four isolates. Thirty-three isolates from multiple serogroups were found to carry the colibactin/pks operon (*clb*), and 23 isolates carried complete *pap* operons, with most isolates carrying at least some *pap* genes (Table S1). All isolates tested negative for carriage of ColV, according to the criteria of Liu et al. (38), indicating that ST1193 isolates are not carriers of these important ExPEC-associated F virulence plasmids. The absence of ColV F plasmids is consistent with the observation that ST1193 does not circulate in food production animals (28).

Antimicrobial resistance genes and class 1 integron structures were widely distributed in ST1193. Most isolates (685 [96.9%]) carry IS*26*, an element strongly associated with the capture and mobilization of antibiotic resistance genes. In total, 467 isolates (66%) carried a class 1 integron. In Fig. 1, BLASTn hit data for *intI1* are presented to highlight integrons that are either truncated or broken due to whole-genome sequencing (WGS) assembly. A complete copy of *intI1* (1,014 bp) was detected in 133 (28.5%) isolates. Notably, 256 (54.8%) isolates carried a 745-bp truncated copy of *intI1* (*intI1*$_{\Delta745}$), which appeared as a BLASTn hit size of 746 bases.

To explore the structures of the most prevalent F plasmids within ST1193 and simultaneously resolve complex resistance structures hosting the *intI1*$_{\Delta745}$ element, we performed comparative analyses of five closed F plasmid sequences. Three hosted the integron under investigation: pAVS0096-a (F⁻:A1:B10*; CP076345.1), as well as pMVC530_A (F⁻:A1:B1; JAMQKE000000000) and pMVC465_A (F⁻:A1:B20; JAMQKF000000000) from strains MVC530 and MVC465, respectively (39). Two additional relevant plasmids were included in this analysis, pNMEC-O75A (ST1193; F⁻:A1:B10; CP030112.1) and pUTI89 (ST95; F29:A⁻:B10; CP000244.1). Figure 2 depicts annotations and BLASTn comparisons of these plasmids. Regarding the complex integron structure, the three resolved examples were variants of the same integron typed by the presence and location of *sul1* and *mphA*, but a Tn*3*-family transposon, which typically mobilizes class 1 integrons (40), was not observed in the annotations. The structure, encompassing the integron and linked resistance gene cargo, appeared to be assembled and mobilized by IS*26*, with a terminal copy of IS*26* truncating *intI1*. The integron hosted *dfrA17* and *aadA5* cassettes. A second tandem IS*26*-associated resistance region

carried *sul2*, *strAB* and *tetAR*, and a third carried $bla_{TEM}$ and *aacC2* in the cases of pMVC530_A and pMVC465_A or $bla_{CTX-M-27}$ in pAVS0096-a. Another closed plasmid from ST1193, pNMEC-O75A, lacked the integron and *sul2* modules, carrying only $bla_{TEM}$ and *aacC2* (Fig. 2). Differences within observed gene counts in the overall ST1193 data set, such as *dfrA17* ($n = 417$) and *aadA5* ($n = 273$), or *sul1* ($n = 287$) and *mphA* ($n = 394$), suggests that the structures we resolved here represent only some of the variation within the complex resistance structures to be found in ST1193.

Outside the aforementioned ARGs, differing integron gene cassette profiles and $bla_{CTX-M}$ alleles were observed both sporadically within the data set and associated with small clusters of isolates. The occasional isolate was also positive for $bla_{CMY}$ ($n = 29$), $bla_{OXA}$ ($n = 62$), and/or *tetB* ($n = 170$). Low counts and multiple alleles of $bla_{IMP}$ ($n = 2$), $bla_{KPC}$ ($n = 11$), $bla_{NDM}$ ($n = 10$), and $bla_{SHV}$ ($n = 1$) were identified (Table S1). Isolates lacking an F plasmid ($n = 34$) also often lacked most ARGs, and similarly, all isolates lacking IS*26* ($n = 22$) also lacked an F plasmid, indicative of links between F plasmid carriage and mobilized multiple-drug-resistance carriage in ST1193. All isolates encoded the *gyrA* fluoroquinolone resistance mutations S83L and D87N. Investigation into $bla_{CTX-M}$ ($n = 325$) showed three primary alleles: CTX-M-27 ($n = 152$), CTX-M-15 ($n = 111$), and CTX-M-55 ($n = 38$). Notably, the latter two alleles were colocalized with complete copies of IS*Ecp1*, while CTX-M-27 was contextualized alongside IS*Ecp1* elements truncated by IS*26*, with IS*26* bordering both ends of investigated contigs. Completed sequences will be required to fully resolve the context of $bla_{CTX-M}$ in ST1193 and to determine if CTX-M-27 is now being mobilized by IS*26* activity, as seemingly observed in pAVS0096-a.

The four *senB*-bearing plasmids depicted in Fig. 2 are representatives of the elements that were widely distributed throughout the ST1193 phylogeny ($n = 606$ [86%]) and in all serogroups. A linear comparison with plasmid pUTI89 is presented in Fig. 2, highlighting that not only is *senB* conserved between the plasmids, but also the entirety of the virulence gene region of the pUTI89 plasmid, as well as a short region of uncharacterized ORFs, is conserved as well. This virulence region is typically split into three sections (i) encoding copper and iron transporters, (ii) encoding iron-associated colicin J sensitivity (*cjrABC*) and carrying *senB*, and (iii) encoding a putative hemin receptor. These regions, including the diverse array of IS elements found within them, were present in all four plasmids, suggesting that F⁻:A1:B1/B10/B20 plasmids are closely related, but with distinct plasmid backbones. In pAVS0096-a and pMVC465_A, it was noted that the region carrying *cjrABC-senB* was reversed between IS*3* family elements. The comparison also highlighted further similarity between F⁻:A1:B1 and F⁻:A1:B10 plasmids, with the novel F⁻:A1:B20 backbone region encoding a type 1 restriction modification system in place of a set of IS elements and VapBC toxin/antitoxin systems encoded by the B1/B10 plasmids.

To observe whether these plasmid assemblies were relevant to the WGS data, pAVS0096-a, pMVC465_A, pMVC530_A, and pUTI89 were aligned against representative F⁻:A1:B1, B10, and B20 plasmid subtypes found in ST1193 whole-genome sequences. A single representative from each of the 14 most prevalent countries of origin was included for each subtype, where available. The BRIG alignments with pUTI89 (Fig. 3A) confirmed the absence of the plasmid transfer region and typically showed full coverage of the virulence regions and their flanking sequence. The alignments demonstrated high conservation of their respective references in B1 and B10 subtype plasmids (Fig. 3B and C), and in B20 isolates overall, but variability across the B20 backbone region (Fig. 3D) was noted. Coverage of the antimicrobial resistance structure was variable, as expected given the promiscuous nature of IS*26*.

To provide additional resolution of the dominant O75:H5 sublineage of ST1193, a phylogenetic tree of all Australian representatives ($n = 243$) was generated (Fig. 4), excluding the distant group highlighted in Fig. 1 to increase average WGS inclusion up to 85% for the analysis (compared to 50% using the global collection against an ST14 reference). The major clustering again split most isolates based on F plasmid subtypes,

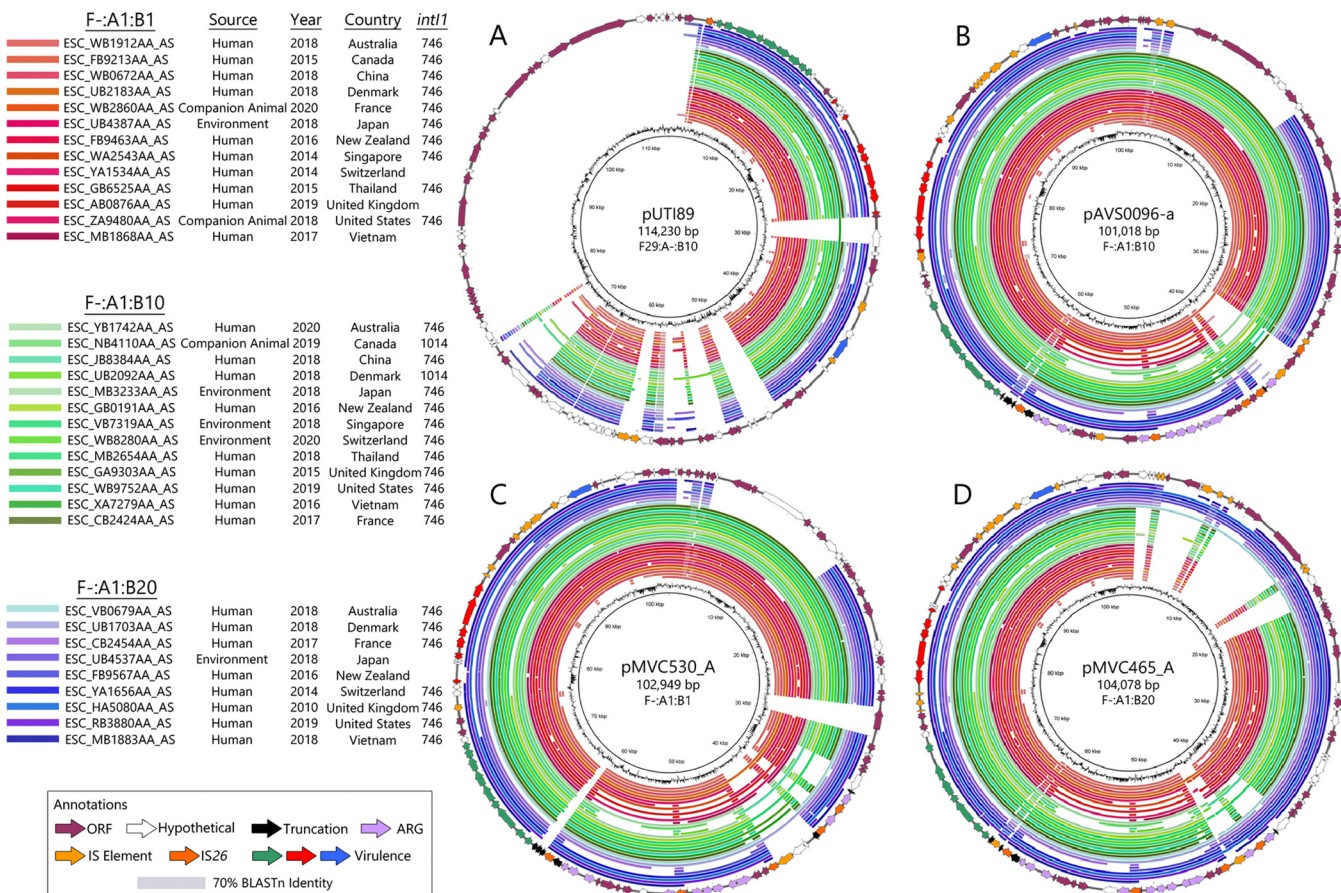

**FIG 3** Circular whole-genome sequence alignments of isolates predicted to host F⁻:A1:B1, F⁻:A1:B10, and F⁻:A1:B20 plasmids against (A) pUTI89, (B) pAVS0096-a, (C) pMVC530_A, and (D) pMVC465_A. Ordering of isolate whole-genome sequences is consistent between the four panels. BLASTn hit range for the alignment rings was set to between 70% and 100% identity. Gene size for *intI1* is shown as the BLASTn hit.

with isolates hosting B20 alleles forming the most distant clade of the data set at approximately 420 SNVs from the reference ESC_FB3738AA_AS. A single isolate typed as B1 was linked to this group. Isolates hosting B10 then formed the primary cluster of isolates at an average of 60 SNVs difference, with three subclades of B1 and one of B20 forming alongside them, ranging from 60 to 90 SNVs. Noticeable clustering, highlighted in Fig. 4, supported One Health associations, with as few as 3 SNVs between the closest pairs of isolates. These connections were observed between humans and companion animals (C1 and C2) or humans and wild avian species (C5, C6, C8, and C10). A pair of isolates obtained 4 years apart (C7) were noted to have truncated *intI1* BLASTn signatures of 682 bp, and a larger clade (C10) had truncated *intI1* hits of 850 bp. Similarly, cluster C9 had some isolates with a truncated *intI1* hit size of 548 bp, another example of which can be seen in C10. Taken together, these data indicate that the lineage has undergone clonal expansion while hosting the *intI1*$_{\Delta745}$ truncation on presumably an F⁻:A1:B10 plasmid, followed by diversification of the lineage, rather than arising through multiple capture events.

A smaller international clade was resolved in a separate phylogeny based around the HierCC H10 cluster 11740 (Fig. 5). This was the second largest H10 cluster in the collection, selected because it had numerous internal H5 groupings and included some of the earliest isolates sequenced. This FIB10 sublineage cluster comprised only human isolates, except one from a wild Australian bird, and the isolates were obtained at significantly different times and locations. Sporadic variations in integrons, AMR profiles, and *senB* carriage were occasionally associated with loss of the dominant F plasmids, indicating again that ST1193 appears to have gone through clonal expansion

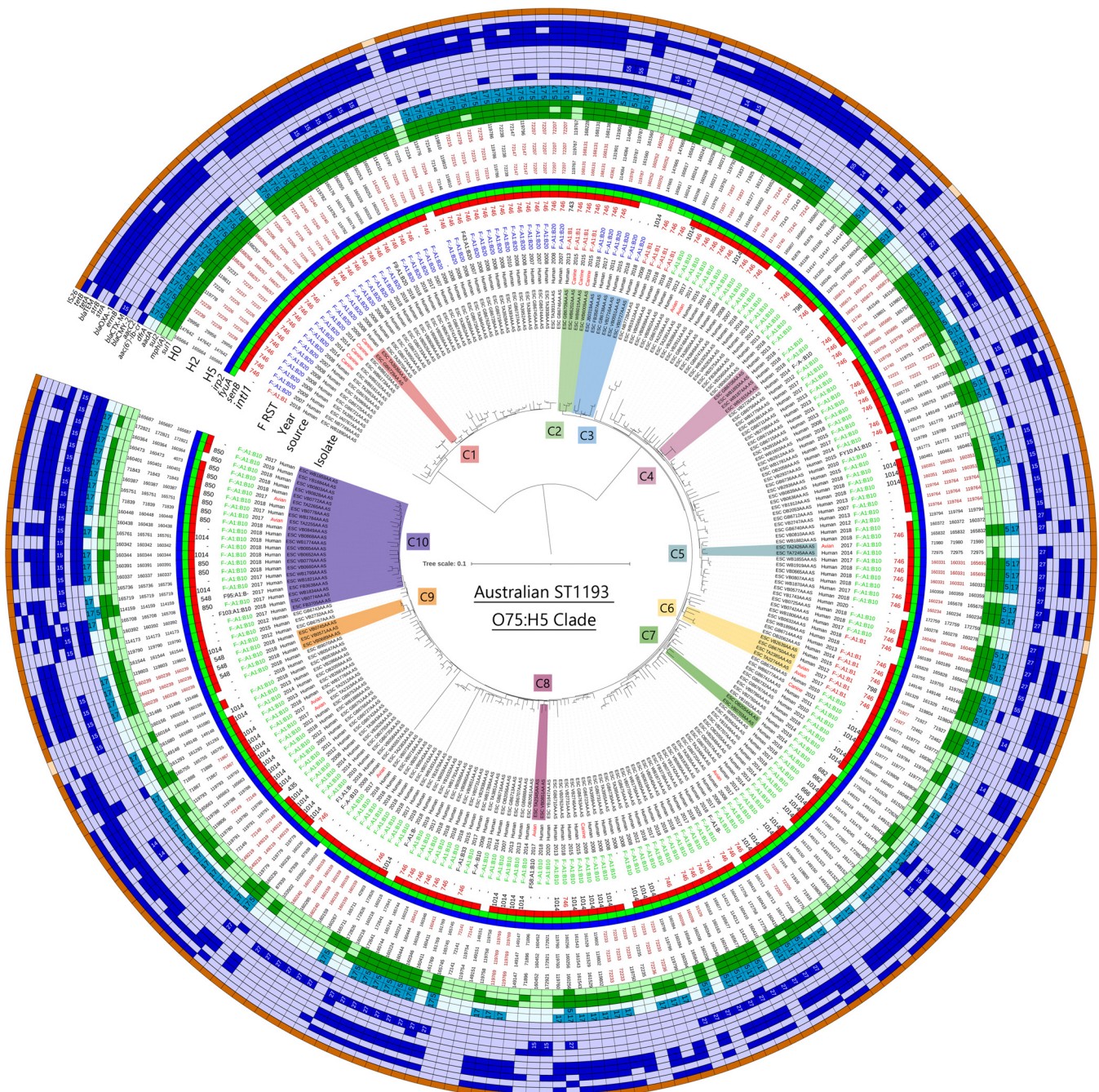

**FIG 4** Phylogenetic core SNV maximum-likelihood tree of the Australian *E. coli* ST1193 O75:H5 clade. The tree is midpoint rooted. The tree scale is in substitutions per site. Sequence ESC_FB3738AA_AS was used as the reference. Metadata and genotypes are presented at the tree tips, with genotypes shown in darker (positive) and lighter (negative) colors on the external rings. Notable clades are also highlighted different colors, with corresponding clade (C) numbers 1 to 10.

mobilizing the *intI1*$_{\Delta745}$ as part of the F plasmid structure, followed by diversification and plasmid loss.

In summary, ST1193 is a pandemic lineage composed of several serogroups, and while there was observable phylogenetic distance between the serogroups, all clades included isolates with virulence plasmids of subtypes F⁻:A1:B1, B10, or B20, with further subgroupings of F plasmid types within the major O75:H5 cluster. These plasmids shared virulence content with pUTI89, a plasmid associated with globally disseminated *E. coli* uropathogens, but with a clear association with specific antimicrobial resistance structures mobilized by IS*26*.

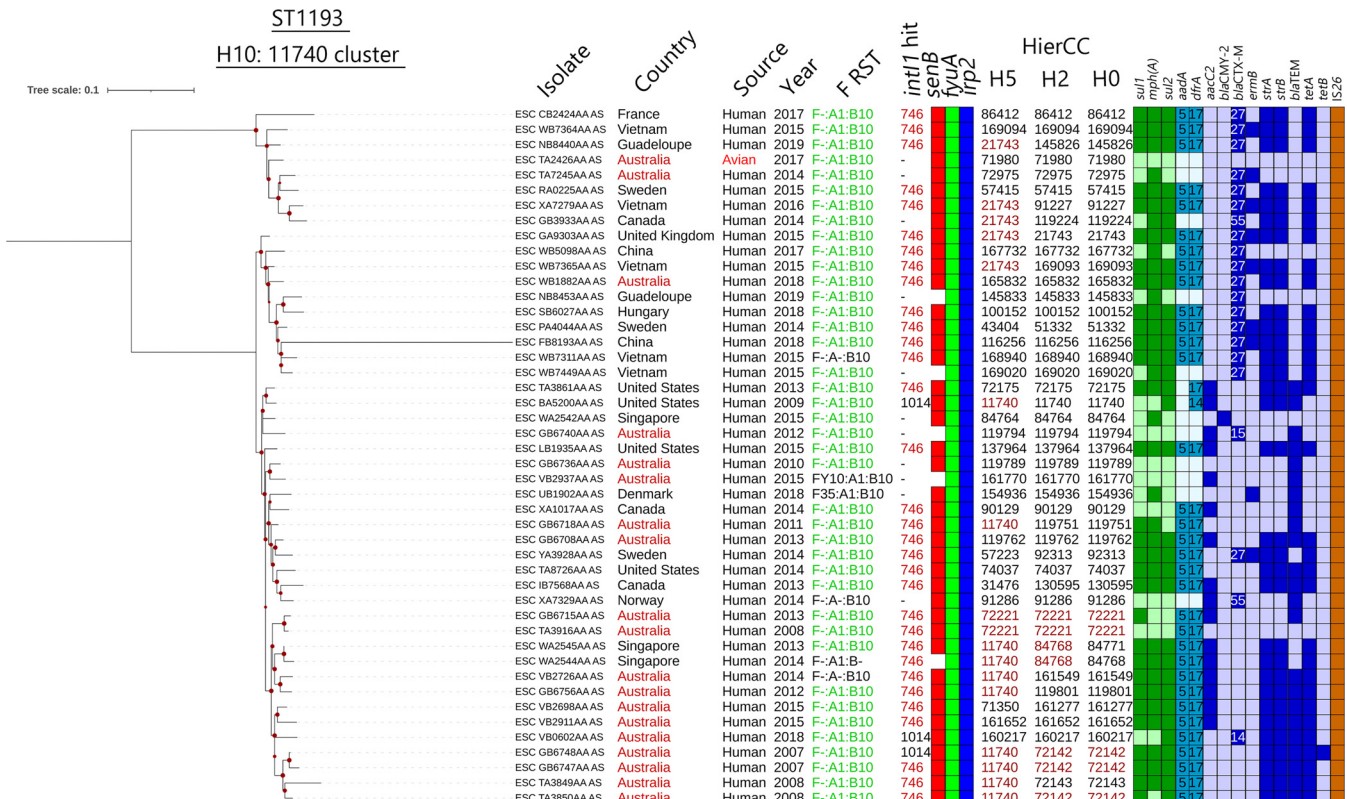

**FIG 5** Phylogenetic core SNV maximum-likelihood tree of *E. coli* ST1193 H10 cluster 11740. The tree is midpoint rooted. The tree scale is in substitutions per site. ESC_BA5200AA_AS was used as the reference. Metadata and genotypes are shown at the tree tips, with genotypes shown in darker (positive) and lighter (negative) colors on the external rings. The *intl1* data are presented as BLASTn hit size.

## DISCUSSION

Here, we describe a detailed phylogenetic analysis of 707 ST1193 genome sequences from 32 countries. Isolates from Australia (259 genomes) and the United States (135 genomes) contributed more than half (394 genomes [56%]) of the collection. The primary source was humans; however, data from companion animals (canine and feline), wastewater, water, and wild birds are indicative of anthropogenic transmission. Isolates from poultry, pigs, cattle, and other food animals are currently absent from public databases. While this may be due to sampling bias, this is a most unlikely scenario. As stated earlier, there are over 44,000 *E. coli* genomes representing 3,737 sequence types sourced from agricultural animals and related sources in EnteroBase, and ST1193 appears only once with a livestock source attribution. We did not find ST1193 in over 250 Australian *E. coli* isolates of porcine origin (41, 42), nor was it detected in *E. coli* from Australian poultry with symptoms of colibacillosis (29). In addition, we have recently reported that *E. coli* isolates that carry pUTI89 (F29:A⁻:B10), the F plasmid replicon sequence type that is frequently found in pandemic ExPEC lineages, are rarely detected among collections of agricultural isolates (28). The weight of evidence supports the contention that ST1193, and *E. coli* isolates hosting F plasmids harboring *cjrABC-senB* in general, are unable to successfully colonize or persist in agricultural animals. The mechanism(s) that underpins host bias is poorly understood, however.

The overwhelming presence of ST1193, often associated with urinary tract infections, in human samples underscored the lineage's pathogenic nature. Understanding the genetic factors that may have helped its success is crucial for identifying future potential ExPEC pathogens. Chromosomal virulence genes were consistently found across the collection, with a high prevalence of the yersiniabactin ferric uptake receptor gene (*fyuA*) and biosynthetic protein gene *irp2*, markers for the *Yersinia* HPI (43), and the *sit* operon. These genes are considered reliable markers of pathogenic potential in *E. coli*

(35). Some other virulence attributes, such as the colibactin and *pap* operons, were also sporadically detected, an indication that the pathogenic potential of the lineage may vary and that despite the overwhelming carriage of key virulence plasmids and the HPI, the presence of other virulence elements may define the importance of specific ST1193 sublineages in the future.

The distribution of phylogenetic clusters within ST1193 matched the different serotypes identified, expanding on earlier observations conducted on a smaller cohort of isolates (2). A similar observation was reported for pandemic ExPEC lineage ST73 (44), highlighting the need for epidemiological investigations beyond the level of resolution that sequence typing can provide. Serotype O75:H5 dominated the ST1193 collection, but O45:H5, O25:H5, O18:H5, and O6:H5 isolates were just as clearly associated with the described virulence and AMR attributes. This suggests that all serogroups descended from a single common ancestor. All serotypes had a H5 flagellar antigen, but a substantial proportion of the collection (*n* = 264 [37%]) were O-nontypeable:H5, again reinforcing the need for further investigation (like HeirCC or phylogenetic analysis) to place these isolates appropriately.

Our observations here on ST1193 expand our understanding of F virulence plasmid variation, host range, and AMR carriage. ST1193 is a significant host of plasmids with similarities to pUTI89, though with heavily altered plasmid backbone sequences (including lost transfer and stability portions in comparison to pUTI89), giving rise to different F RSTs which are with association to *cjrABC-senB* (2). The *senB*-encoding plasmids pAVS0096-a (F⁻:A1:B10), pMVC465 (F⁻:A1:B20), and pMVC530 (F⁻:A1:B1) are representatives of plasmids that are widely distributed throughout the ST1193 phylogeny and are hosted in all serogroups within the lineage, indicating that it was carried by the progenitor strain of the lineage. Comparisons with pUTI89 (F29:A⁻:B10) showed the virulence gene region and surrounding genetic environment are highly conserved in all plasmids, including mobile elements. However, unlike pUTI89 (F29:A⁻:B10), which often carries a single copy of IS*26*, these plasmids carry a complex resistance gene region (CRR) and often have a specific resistance gene structure containing a class 1 integron with a truncated *intI1*, mobilized solely by IS*26*. These observations led us to hypothesize that pAVS0096-a (F⁻:A1:B10), pMVC465 (F⁻:A1:B20), and pMVC530 (F1:A1:B1) are potential relatives or derivatives of pUTI89, but with distinct plasmid backbones, that have emerged within ST1193.

Similar to the pandemic lineages ST127 (45), ST95 (28, 46), and ST131 (32), which carry pUTI89, ST1193 isolates carry *senB*⁺ plasmids and are not found in poultry, swine, or cattle but rather are widely found in humans, dogs, cats, wastewater, and wildlife, particularly urban birds. Furthermore, all plasmids under analysis carried a copy of IS*26* in the same position, abutting copper sensitivity suppression open reading frames (ORFs), although this position appears to be variable due to repeated IS*26* activity. In addition, alignment of the four plasmid sequences showed that pAVS0096-a (F⁻:A1:B10), pMVC465 (F⁻:A1:B20), and pMVC530 (F1:A1:B1) share considerable sequence identity with pUTI89 but that the entire transfer (*tra*) module in pUTI89 is missing from the F plasmids found in ST1193. It is conceivable that the loss of the *tra* region in these ST1193 F virulence plasmids enabled the acquisition and stable integration of the CRR into the plasmid backbone. Stephens et al. (46) hypothesized that pUTI89, possibly in concert with chromosomal genes, may inhibit lineages of ST95 from acquiring other plasmid cargo, limiting their ability to acquire antimicrobial resistance. Consistent with this hypothesis, extensive phylogenomic analyses of ST95 (*n* = 668) showed that lineages that carry pUTI89 (F29:A⁻:B10) are pan-sensitive and lack evidence of class 1 integrons (28). It is not known if the F plasmids in ST1193 do the same, although carriage of additional plasmids in ST1193 isolates was noted, with the second most prominent plasmid subtype, I1, detected in 17% of isolates. These observations suggest that the genetic modifications found in the F plasmids hosted by ST1193 may enable the lineage to acquire other plasmid families. Further studies should decipher these evolutionary developments and seek evidence of these plasmids in other *E. coli* lineages, as

they encode combinations of important virulence and antibiotic resistance gene cargo and pose an elevated threat to human health.

Distribution of ARGs within the data set was quite variable, but some key genotypes alongside truncations of *intl1*, caused by IS*26*, served as prominent markers that highlight the types of complex resistance structures found within ST1193. The key feature of the most prevalent integron, identified by the conserved 745-bp truncation of *intl1*, was that it was not associated with Tn*21*, the classical mobilizer of class 1 integrons associated with *sul1* and *mphA* (40), and were instead mobilized by IS*26* (47, 48). The variable presence of ARGs suggests that plasmids carried by ST1193 often undergo alterations to their ARG cargo, likely mediated by IS*26*. The IS*26*-mediated deletion event that featured prominently in ST1193 is likely to have rendered the integrons unable to capture, lose, or rearrange gene cassettes, while retaining their gene cassette expression functionality. This results in the "locking in" of gene cassettes in these truncated integron structures and their retention in place until they are activated by a functional *intl1* present elsewhere in the genome or lost via homologous recombination. The $intl1_{\Delta745}$ truncation has been reported previously in wild Australian birds, albeit rarely, notably appearing in association with F29:A⁻:B10 and F2:A⁻:B10 (both pUTI89-like) plasmids within *E. coli* ST131 and ST10 isolates in 2012 (33). This same study also reported $intl1_{\Delta682}$ BLASTn signatures from ST624, ST457, and occasionally other STs. This truncation size was also observed here in ST1193, similarly associated with carriage of *dfrA17* and *aadA5*. These associations suggest that this integron structure is in circulation within ExPEC outside the F plasmids found within ST1193.

The global dissemination and clinical association of these IS*26* mobilized integrons is of major concern and highlights the rapid adaptability of these urinary tract pathogens during antimicrobial treatment or, importantly, in synanthropic species, where exposure to selection pressures is more indirect. These developments demonstrate the next stage in the evolution of class 1 integrons and the resistance gene cargo they carry (33). Some other ARGs, $bla_{CTX-M}$ alleles being a key example, had a distribution disconnected from that of the class 1 integron, indicating that they were mobilized separately in ST1193. It is notable that chromosomal integration of $bla_{CTX-M}$ genes is on the rise in *E. coli* (49), including clinically relevant *E. coli* lineages that carry pUTI89, such as ST963 (50) and ST131 (32).

Multiple-drug resistance in *E. coli* that cause extraintestinal disease can lead to increased morbidity (extended hospital stays) and mortality and has a dominant influence on the epidemiology and management of infectious disease. *E. coli* ST1193 has spread globally at a high rate, given that the earliest entry of ST1193 in EnteroBase is 2007 and it was absent, for example, in 403 WGS of human commensal *E. coli* isolates collected between 1980 and 2010 (23). Understanding virulence, however, is complex because of the redundancy in the myriad of mobile elements that carry virulence gene cargo, especially in ExPEC. The number of ST1193 isolates recovered from human clinical cases serves as a strong indicator of its importance as a pandemic ExPEC lineage. Our study highlights the urgency of better understanding ExPEC lineages and genotype distributions, as well as identifying markers to differentiate and track future variation.

**Conclusion.** Members of the globally disseminated pandemic lineage ST1193 carry F plasmids encoding the pUTI89 virulence region comprising the copper and iron transporters, *cjrABC* and *senB*, and a putative hemin receptor but appear to differ markedly from pUTI89 in their capacity to be infiltrated by complex resistance regions assembled by the activity of IS*26*. These F plasmids, which appear to have developed within ST1193, are potential derivatives of pUTI89 but with smaller, distinct plasmid backbones and consistent carriage of ARGs. Our data contribute to the hypothesis that *senB*-carrying F plasmids are found in ExPEC lineages that colonize humans or have close human association, like companion animals and urban wildlife.

## MATERIALS AND METHODS

**Long-read whole-genome sequencing.** The isolation and short-read sequencing of *Escherichia coli* isolates MVC465 and MVC530 from Australian canine companion animals was described previously (39). Both were isolated at the University of Melbourne Veterinary Hospital; MVC465 was taken from a canine

urine sample in 2014, and MVC530 was obtained from urine through cystocentesis in 2015. Whole-genome sequencing of both isolates using a long-read platform was performed as follows. Genomic DNAs were extracted from overnight cultures using the Promega Wizard HMW kit. Sequencing libraries were prepared from approximately 400 ng genomic DNA using the Oxford Nanopore rapid barcoding kit SQK-RBK004. The libraries were loaded onto a FLO-MIN106 R9.4 flow cell fitted on a MinION Mk1B device. Live base calling of the MVC465 and the MVC530 raw reads was performed by MinKNOW versions 4.2.5 and 4.5.4, respectively, running Guppy versions 4.3.4 and 5.1.15, respectively, with the "fast" R9.4.1 model. Demultiplexing and barcode trimming were performed on passed reads using Guppy barcoder version 5.1.15. The resulting fastq files were filtered for quality scores of >8 and lengths of >1,000 using NanoFilt (51). Completed assemblies were generated with Unicycler v0.5.0 using both short and long sequence reads.

**Data acquisition.** The whole-genome sequences, isolation metadata, e-serotypes and HierCC (multilevel cluster assignment) (52) values for *E. coli* ST1193 (*n* = 707) were downloaded from EnteroBase (53) (https://enterobase.warwick.ac.uk/) on 20 August 2021. EnteroBase entries that did not include isolate source, year, and/or country of origin were excluded. Plasmid sequences were sourced from the GenBank nucleotide database (https://www.ncbi.nlm.nih.gov/genbank/).

**Genotyping and phylogenomic analyses.** Virulence genes, plasmid replicons, and mobilized antimicrobial resistance genes were detected using ABRicate (https://github.com/tseemann/abricate) and the databases VirulenceFinder (54), PlasmidFinder (31) and ResFinder (55), respectively. ABRicate was also used for F plasmid subtyping using the F RST scheme (available at https://pubmlst.org/organisms/plasmid-mlst). ABRicate was run using default settings outside custom database selection. The class 1 integrase gene *intI1*, IS*26*, IS*Ecp1*, *gyrA* mutations, and the colibactin operon *clb* were identified separately using BLASTn (56) with no cutoff values set and then manually curated to 90% identity and differing coverage values depending on observed truncations. Phylogenetic trees and SNV counts were generated using Parsnp (with -x and -c flags) and gingr v1.2 from the Harvest Suite (57) (https://github.com/marbl/parsnp) and visualized in iTOL (58) (https://itol.embl.de/). Genomic alignments for comparative analysis of sequence data were performed using progressiveMauve (59). Circular BLASTn alignments were generated using BRIG (60), and linear alignments with Easyfig (61). Annotations were generated with RASTtk (62). SnapGene was used to handle and output sequence annotations. Typing data were cross-referenced with outputs from the Centre for Genomic Epidemiology (http://www.genomicepidemiology.org/services/).

**Data availability.** Data generated for this project can be found under GenBank BioProject accession PRJNA846578.

## SUPPLEMENTAL MATERIAL

Supplemental material is available online only.
**SUPPLEMENTAL FILE 1**, XLSX file, 0.2 MB.
**SUPPLEMENTAL FILE 2**, XLSX file, 5.9 MB.
**SUPPLEMENTAL FILE 3**, PDF file, 0.1 MB.

## ACKNOWLEDGMENT

This project was partly funded by the Australian Centre for Genomic Epidemiological Microbiology (AusGEM), a collaborative partnership between the NSW Department of Primary Industries and the University of Technology Sydney. These results are helping to inform the OUTBREAK decision support system, funded by the Medical Research Future Fund Frontier Health and Medical Research Program (MRFF75873).

E.R.W. designed and performed data analysis and drafted and edited the manuscript. R.N.B. performed bacterial isolation and identification. M.S.M. generated genome sequences and edited the manuscript. G.F.B. assisted with analysis design and edited the manuscript. S.P.D. designed analysis, drafted and edited the manuscript, acquired funding, and provided supervision.

We declare that we have no competing interests.

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
