## [Reviewer comments · Microbiology Spectrum]

Microbiology Spectrum

Global phylogeny and F virulence plasmid carriage in pandemic *Escherichia coli* ST1193

Ethan Wyrsh, Rhys Bushell, Marc Marena, Glenn Browning, and Steven Djordjevic

Corresponding Author(s): Steven Djordjevic, Australian Institute for Microbiology and Infection

Review Timeline:

Submission Date:	July 4, 2022
Editorial Decision:	August 12, 2022
Revision Received:	September 20, 2022
Editorial Decision:	October 16, 2022
Revision Received:	November 1, 2022
Accepted:	November 3, 2022

Editor: Cheryl Andam

Reviewer(s): The reviewers have opted to remain anonymous.

Transaction Report:

DOI: <https://doi.org/10.1128/spectrum.02554-22>

August 12, 2022

Prof. Steven Philip Djordjevic
Australian Institute for Microbiology and Infection
15 Broadway, University of Technology Sydney
Ultimo, NSW 2007
Australia

Re: Spectrum02554-22 (Global phylogeny and F virulence plasmid carriage in pandemic Escherichia coli ST1193)

Dear Prof. Steven Philip Djordjevic:

Editor's comments: Both reviewers have major concerns about the lack of details in the Methods, lack of rationale/importance for studying ST 1193, and that the Conclusions do not support the results and data presented. Some of the figures are also not legible to read.

Link Not Available

Sincerely,

Cheryl Andam

Journals Department
Reviewer comments:

Reviewer #1 (Comments for the Author):

Wyrsh et al. compared ST1193 genomes using mostly the database data with focuses on F plasmids. The also completed 4 ST1193 genomes in this study. The purpose of the study and results are of interest and will contribute to understanding of ST1193. The methods used look up-to-date. However, the manuscript needs substantial revision in scientific writing. Many descriptions are obscure without objective data or clear presentation and look like lacking scientific validity. Some of them were indicated below.

Why only two isolates underwent long-read sequencing in this study? If they are important, their characteristics should be described in a separate section.

Please clarify, how the authors performed plasmid analysis using WGS data.

Line 10-11. These STs might not be needed here considering the focus of the study.

Line 15. "707 isolates" Please briefly describe what these isolates were. Clinical isolates? What kind of infection? Resistant to antimicrobials? Which strains did the authors obtain and which were from databases? Methods are not adequately described. What methods were used for molecular investigation?

Line 21-22. This sentence looks not describing study results.

Line 23-24. Association with AMR carriage has not been described.

Line 27-34. Importance needs complete rewriting as this only describes detailed results of the study, not importance or interpretation of the study that is based on the results obtained.

Line 28. Please elaborate with RST.

Line 38-69. Backgrounds explaining F plasmids are needed, if the authors focus F plasmids (as described in the title) in this study.

Line 73-74. Need a brief description for these two isolates.

Line 85. Please elaborate with "HierCC".

Line 89. As the authors described in Introduction, fluoroquinolone resistance genes (especially QRDRs) are important for the ST1193. This should be incorporated into analysis.

Line 90. "Most genes" Does this mean, genes other than int1/IS26 were identified by these databases?

Line 94-96. This can be described in Results.

Line 107. Why only class 1 integron and IS26 were included for the analysis? Please provide explanations in Introduction.

Line 113. How overwhelming? Please provide data. "Many isolates were O-typeable" (line 114) looks contradictory. How many?

Figure 1. Image resolution of Figure 1 is too low to illustrate the exact figure. I could not see branch lines and characters. It is difficult to review this figure. Which ST14 strain was used for root? What does O75:H5 (Distant group) mean?

Line 117-118. Clusters formed by these serotypes are not visible on Figure 1. Individual O:H types can be converted to color bars. I think Figure 1 should be simplified to limit important results here, like O:H types, plasmid types, source, and countries. Only key VAGs or ARGs are needed.

Line 119-122. Same as above for F plasmid types. Authors should present supporting data for these sentences (percentage of O75:H5 isolates which had the specific F plasmid, etc.).

Line 123. "A lack of F plasmid diversity" this can be moved to Discussion and it must be accompanied with scientific data showing less diversity of ST1193 F plasmids compared with other clones.

Line 130-131. How many assembled plasmids? Did the authors mean "closed plasmids"?

Line 131-132. It is hard to believe as many as 591 isolates had senB and F plasmid replicons on the same contig.

Line 132-133. Please provide data for senB prevalence in E. coli data in Enterobase.

Line 143-146. These interpretations are discussions.

Line 157. Some isolates have more than one class 1 integrons. How were these managed? Isolates have "broken" contigs that had int1 sequence in the end of contigs and these can be recognized as truncated int1 if the authors do not limit the analysis to closed genomes or plasmids.

Line 162. What is "these plasmids"? Plasmids were not described in this paragraph and int1 can be on chromosomes. Please describe number of closed plasmids first and then number of isolates with the genes of interests.

Line 162. Was detection method for Tn3 transposon described in Methods?

Line 159. Please elaborate with "specific resistance structure".

Line 165-168. Please revise these sentences. It is hard to understand what was second, third, and fourth and why specific plasmids were described with number of isolates positive with the genes.

Line 172-172. Provide exact data for these descriptions.

Line 181-182. Please describe why these 4 plasmids were representative of ST1193 plasmids, i.e., percentages of isolates carrying these plasmids.

Line 182. How dominant? How many isolates were associated with deletion?

Figure 2. Please indicate plasmid backbone genes (such as rep and mob) to clarify differences and similarities among these plasmids.

Line 189. "distinct plasmid backbones" Please provide rationale for this.

Line 193-195. If the FIB allele is not identical to FIB10, this plasmid did not belong to B10 plasmid. If it is a new allele, assign the new allele number and discuss.

Line 196-197. Four plasmids were aligned with 14 representative plasmids. Were these 14 plasmids closed? How were these 14 selected? Please provide details (size, pMLST, etc) of these 14 plasmids.

Reviewer #3 (Comments for the Author):

The manuscript reports phylogenetic analysis of 707 E. coli ST1193 genome sequence from 32 countries retrieved from Enterobase and analysed together with long read genomes sequences of two E. coli strains for plasmid characterisation. The authors describe the distribution of E. coli ST1193, plasmid pUTI89, class 1 integrase gene int1 and the insertion sequence

IS26 across host organisms. They conclude that "senB-encoding F plasmids are found in ExPEC lineages that colonise humans or have close human association, such as companion animals and urbanised wildlife, but not E. coli that colonise poultry, pigs, and ruminants."

The manuscript is well written and easy to comprehend. My main concern however, is that the study conclusion is baseless as there is no data, presented in the paper or previously published, to support it. Additionally, whilst it is appreciated that ST1193 is an important pathogenic E. coli ST, the authors have not articulated, why this study is important.

1. L22: "Absent food animals" should be "absent in food animals"
2. L26-35: The importance of the study is unclear. The authors give a background on the distribution of F plasmids and a brief description of pUTI89. At L34-35 the authors state that "Our data suggests that senB+ plasmids are found in ExPEC lineages that colonise humans and companion animals but not E. coli that colonise food animals. " First, on reading the whole manuscript, this sentence is wrong as it is not backed by data. Seccon, even if it were the sentence were correct, there is nothing to show why this result and indeed the whole study are important, which what this section is about.
3. L28: Define RST on first time use
4. L50: What evidence is there to suggest that ST1193 has never been recovered from poultry, pigs or ruminants? Did the authors review all ST1193 papers and found no reports of ST1193 from these animals? Even if no report of ST1193 in these animals was found, there has no evidence of sampling from such animals. None of the papers cited at L49 had ever attempted to sample poultry, pigs or ruminants. The lack of report of ST1193 from these animals may not necessary imply that these animals don't host ST1193, but rather, a result of sampling bias.
5. L90-93
 - a. "Most genes were detected using ABRicate...". What tool was used to detect "other (fewer)" genes?
 - b. Authors should also specify what genes were detected with ABRicate. Of course, from the mentioned databases [L91], what can infer that these were virulence, plasmid replicon and AMR genes. However, this may not be obvious to all readers and the authors should explicitly state the gene types they searched for.
 - c. At L91, the authors mention ResFinder as one of the databases they used to search for genes using ABRicate. At L92-93, the author state that "ABRicate was run using default settings". The default database for ABRicate is the ncbi amr database. Can the authors clarify on which database they exactly used? If indeed the authors used ResFinder, what was their reason for not using the default (NCBI) database, which has more non-redundant AMR gene variant sequences than ResFinder?
6. L93-94 and L203: If BLASTn was run without cut-off values, what was the threshold for determining presence or absence of class 1 integrase gene int11, int11 truncations, and IS26? And how consistent is this with 208-209 where identity match of 70-100% are specified?
7. L116: "Phylogenetic distribution" should be "Phylogenetic tree"
8. L123: "Limited plasmid diversity" would be more appropriate than "lack of plasmid diversity" as there is some diversity.
9. L226-L229 and L239: Include colour keys for Figure 4 and Figure 5 or describe what the colours represent in the respective figure legends.
10. L253-258: The absence of isolates from pigs cattle and other food animals is clearly due to sampling bias. The cited studies in this paper, which also contributed to the ST1193 genomes obtained from Enterobase for this study never attempted to sample from poultry, pigs or ruminants and thus as in my previous comment, the absence of genomes of isolates from these animals does not imply the animals don't carry E. coli ST1193.
11. L344-346: "Our data contribute to the hypothesis that senB-encoding F plasmids are found in ExPEC lineages that colonise humans or have close human association, such as companion animals and urbanised wildlife, but not E. coli that colonise poultry, pigs, and ruminants". This study has no data on poultry, pigs and ruminants hence has no basis to make the such a conclusion. L344-L346, should be deleted or amended to reflect the data reported in the manuscript.

Staff Comments:

Preparing Revision Guidelines

Please return the manuscript within 60 days; if you cannot complete the modification within this time period, please contact me. If you do not wish to modify the manuscript and prefer to submit it to another journal, please notify me of your decision immediately so that the manuscript may be formally withdrawn from consideration by Microbiology Spectrum.

Aug 2022 review of global phylogeny Paper on F factor in *E. coli* ST1193 for Microbiology Spectrum Review.

The paper is on the phylogeny the F plasmids in a single important lineage of *E. coli*. This is not commonly done and reveals the complexity in this common plasmid group and is a valuable contribution, but I do have suggestions for presentation.

1/ I suggest the introduction include a brief section on F plasmids and their distribution and cite a few papers as background, such as “Genomic network analysis of environmental and livestock F-type plasmid populations” by Matlock et al (ISME Journal (2021) 15:2322–2335 <https://doi.org/10.1038/s41396-021-00926-w>)

2/ L10 lists 6 STs responsible for about half of the human EPEC, and then refers to major subject ST1193. Suggest put ST1193 details here with comment on where it fits into the human EPEC numbers.

3/ There is no easily read tree figure of ST1193 to show the overall branching pattern, as the tree presented in Figs 1 and 4 is in a circular form, in which the relationships of the clades is not easily seen in Fig 1, with less problem for Fig 4. The line is too thin in Fig1, which can be fixed, but the circular format of the tree makes it difficult to read,

4/ I cannot read the text in Fig1 because on enlargement the characters are too blurred. I suggest increase resolution to allow the details to be read on amplification, and also add a standard tree figure of clade relations as shown for the H10 isolates in fig 5. There are also many faint circular lines in Fig 1 that should not be there at all.

The basic problem is that Fig 1 has far too much data for details to be resolved on an A4 page, but the resolution should be such that amplifying the page gives readable text. It may be best to also give the data in a form that allows those interested to generate traditional tree figures of all or parts of the data.

5/ Fig 4 (L226) has the same problems but less serious as less data. Fig 5 shows the advantage of a traditional tree as the tree is clear and data all readable.

6/ However, while Figs 1 and 4 are mostly text, the text is not able to be highlighted or searched, and I suggest the data should also be presented in a table.

7/ L12, does “their” refer to refer to just pUT189; if so I suggest use “its”. Also the statement on role of plasmid pUT189 on host range and resistance needs a reference cite.

8/ I suggest give meaning of “HierCC” “intl1 hits”.

9/ L147 Table 1 appears to be a list of all isolates, but all are H5, while elsewhere other H antigens are given with Fig 5 being a tree of an H10 cluster. Clarification is needed

10/ L15. Are the “clusters” referred to also clades. If so I suggest call them clades for consistency.

11/ L27 – as for L12. This statement also needs a reference(s).

12/ L42 – It seems to me that it is not the recombination event that is remarkable but the subsequent events.

13/ L85 and L237 fig 5 The paper on the HierCC clustering protocol should be cited • DOI: [10.1093/bioinformatics/btab234](https://doi.org/10.1093/bioinformatics/btab234)

14/ L123 et seq. The lack of diversity is said to be a key feature of ST1193, but there is no benchmark for diversity. It is not obvious what level of diversity is expected and thus making lack of diversity a key feature.

15/ L135 .I am surprised that 37% cases in Table have O antigen not typable (L267), particularly in Australia where it is shown as 50% (table 1). There is a long history of defining O antigens and currently 187 structurally different form are recognised are recognised and distinguished by serology (Liu et al 2019). It is generally rare for an *E. coli* isolate to not have one of those identifiable O antigen sequence at the Oag locus, and other studies based on all available genome sequences have not reported a high number of untypables.. An example is the 2016 paper "In silico serotyping of *E. coli* from short read data identifies limited novel O-loci but extensive diversity of O:H serotype combinations within and between pathogenic lineages (doi: 10.1099/mgen.0.000064. eCollection)

16/ I suggest that further analysis be undertaken as I suspect that all or most O antigens in this study could be determined.

17/ L148 and L225. Figs 1 and 4 have a lot of text but it is not legible even on amplification so needs higher resolution, The data should also be capable of cut and paste for detailed examination, which probably means a copy in a file format in the SI.

18/ L176 – Fig 2. Is there anything of significance to say about the relationships of the 4 plasmids with plasmid UT189. A key feature in that Figure is that one region (coloured red) can be in either orientation, and as shown the plasmid order is such that the orientation alternates from top to bottom. Can that order be changed to make 2 blocks, one for each orientation, which would make that more obvious. However it may be that there is meaning in the current order that I overlooked.

To the editor and reviewers,

We would like to thank everyone involved in the review process, your queries and suggested edits have helped us greatly improve the manuscript. We have attempted to answer and make changes relating to most queries and have otherwise performed significant editing to the manuscript as detailed by the track changes. This includes adding new data and completely re-writing some sections, and as such some queries are no longer relevant, which we have noted in our individual responses. Please note also that line numbers have changed significantly and are often no longer relevant, so they have been excluded from our responses.

We hope you enjoy our updated version and thank you once again for your consideration.

Regards,

Prof. Steven Djordjevic

Reviewer comments:

Reviewer #1 (Comments for the Author):

Wyrsh et al. compared ST1193 genomes using mostly the database data with focuses on F plasmids. The also completed 4 ST1193 genomes in this study. The purpose of the study and results are of interest and will contribute to understanding of ST1193. The methods used look up-to-date. However, the manuscript needs substantial revision in scientific writing. Many descriptions are obscure without objective data or clear presentation and look like lacking scientific validity. Some of them were indicated below.

Why only two isolates underwent long-read sequencing in this study? If they are important, their characteristics should be described in a separate section.

Please clarify, how the authors performed plasmid analysis using WGS data.

The isolates were sequenced specifically to resolve the F plasmid content. We hope this has been clarified in the updated manuscript. Similarly, descriptions of WGS comparisons to plasmid sequences are now more clearly defined.

Line 10-11. These STs might not be needed here considering the focus of the study.

Removed.

Line 15. "707 isolates" Please briefly describe what these isolates were. Clinical isolates? What kind of infection? Resistant to antimicrobials? Which strains did the authors obtain and which were from databases? Methods are not adequately described. What methods were used for molecular investigation?

Abstract has been heavily rewritten, including the addition of requested details.

Line 21-22. This sentence looks not describing study results.

As above.

Line 23-24. Association with AMR carriage has not been described.

As above.

Line 27-34. Importance needs complete rewriting as this only describes detailed results of the study, not importance or interpretation of the study that is based on the results obtained.

Importance has been re-written.

Line 28. Please elaborate with RST.

Has been more clearly defined and referenced within the introduction.

Line 38-69. Backgrounds explaining F plasmids are needed, if the authors focus F plasmids (as described in the title) in this study.

A section introducing F plasmids has been added.

Line 73-74. Need a brief description for these two isolates.

Description of isolates has been added.

Line 85. Please elaborate with "HierCC".

A basic definition and reference have been added.

Line 89. As the authors described in Introduction, fluoroquinolone resistance genes (especially QRDRs) are important for the ST1193. This should be incorporated into analysis.

An analysis was performed to confirm the presence of key QRDR mutations.

Line 90. "Most genes" Does this mean, genes other than *int11*/IS26 were identified by these databases?

More specific language has been used to clarify which genes are meant, including additional genes included with the new analysis.

Line 94-96. This can be described in Results.

Has been moved to results.

Line 107. Why only class 1 integron and IS26 were included for the analysis? Please provide explanations in Introduction.

The importance of *int11* and IS26 has been added to the introduction.

Line 113. How overwhelming? Please provide data. "Many isolates were O-typeable" (line 114) looks contradictory. How many?

This point was revised, and a more appropriate explanation provided in relation to data as presented in Figure 1.

Figure 1. Image resolution of Figure 1 is too low to illustrate the exact figure. I could not see branch lines and characters. It is difficult to review this figure. Which ST14 strain was used for root? What does O75:H5 (Distant group) mean?

Hopefully upon second review, a high-quality image will be available. Requested details have been added to the figure legend.

Line117-118. Clusters formed by these serotypes are not visible on Figure 1. Individual O:H types can be converted to color bars. I think Figure 1 should be simplified to limit important results here, like O:H types, plasmid types, source, and countries. Only key VAGs or ARGs are needed.

While we appreciate the sentiment and agree that there is a lot of data presented, however large genome analyses require comprehensive overlapping datasets to draw meaningful conclusions- we have elected to keep the figure and hope a higher resolution figure will be sufficient to convince you of its importance.

Line 119-122. Same as above for F plasmid types. Authors should present supporting data for these sentences (percentage of O75:H5 isolates which had the specific F plasmid, etc.).

Numbers have been added to support all statements.

Line 123. "A lack of F plasmid diversity" this can be moved to Discussion and it must be accompanied with scientific data showing less diversity of ST1193 F plasmids compared with other clones.

Has been removed for clarity, because F plasmid diversity is only truly known for a small number of sequence types. A more comprehensive study will be required to properly explore this observation.

Line 130-131. How many assembled plasmids? Did the authors mean "closed plasmids"?

This has been clarified in the description of plasmid comparisons.

Line 131-132. It is hard to believe as many as 591 isolates had *senB* and F plasmid replicons on the same contig.

We apologise for the poorly worded inference, clearer language is now used – we are only stating that 591 isolates were positive for gene *senB*.

Line 132-133. Please provide data for *senB* prevalence in *E. coli* data in Enterobase.

Unfortunately, that data requires significant characterisation and explanation that does not yet exist yet in a significant way in current literature, so this has been removed for clarity. A future study will be required to properly explore this data.

Line 143-146. These interpretations are discussions.

Both data and discussion points surrounding the observation that ST1193 is not found in animal agriculture have been included at various points.

Line 157. Some isolates have more than one class 1 integrons. How were these managed? Isolates have "broken" contigs that had int1 sequence in the end of contigs and these can be recognized as truncated int1 if the authors do not limit the analysis to closed genomes or plasmids.

We have ensured the use of a specific term "BLASTn hits" rather than truncation sizes regarding this data, for this reason. More detail regarding this has been added to ensure readers understand the context of the data, which we believe is useful for analysis even without inference and curation based on contig positioning.

Line 162. What is "these plasmids"? Plasmids were not described in this paragraph and int1 can be on chromosomes. Please describe number of closed plasmids first and then number of isolates with the genes of interests.

This section has been re-written, and while we find the gene counts interesting and indicative of variation in the collection, they have been removed to keep focus on the plasmids at this point in the manuscript.

Line 162. Was detection method for Tn3 transposon described in Methods?

Merely a comment based on annotation and our knowledge of the "usual" integron structure, which has been made clearer.

Line 159. Please elaborate with "specific resistance structure".

Changed within re-write of section.

Line 165-168. Please revise these sentences. It is hard to understand what was second, third, and forth and why specific plasmids were described with number of isolates positive with the genes.

Section has undergone a significant re-write for clarity.

Line 172-172. Provide exact data for these descriptions.

Provided.

Line 181-182. Please describe why these 4 plasmids were representative of ST1193 plasmids, i.e., percentages of isolates carrying these plasmids.

Provided.

Line 182. How dominant? How many isolates were associated with deletion?

Provided.

Figure 2. Please indicate plasmid backbone genes (such as rep and mob) to clarify differences and similarities among these plasmids.

Additional annotations have been added to Figure 2.

Line 189. "distinct plasmid backbones" Please provide rationale for this.

Additional annotation and cleared description have been provided.

Line 193-195. If the FIB allele is not identical to FIB10, this plasmid did not belong to B10 plasmid. If it is a new allele, assign the new allele number and discuss.

Upon re-analysis, it was found that these alleles were mistyped due to contig breaks and/or plasmid circularisation positioning at the *rep* gene being typed, leading to breaks in the BLASTn hit that was called as a mistype. Reference to this has now been removed.

Line 196-197. Four plasmids were aligned with 14 representative plasmids. Were these 14 plasmids closed? How were these 14 selected? Please provide details (size, pMLST, etc) of these 14 plasmids.

This data was of representative WGS predicted to have the specified F plasmid types. Changes in the text and in the Figure 3 description highlights this.

Reviewer #3 (Comments for the Author):

The manuscript reports phylogenetic analysis of 707 E. coli ST1193 genome sequence from 32 countries retrieved from Enterobase and analysed together with long read genomes sequences of two E. coli strains for plasmid characterisation. The authors describe the distribution of E. coli ST1193, plasmid pUTI89, class 1 integrase gene *intI1* and the insertion sequence IS26 across host organisms. They conclude that "senB-encoding F plasmids are found in ExPEC lineages that colonise humans or have close human association, such as companion animals and urbanised wildlife, but not E. coli that colonise poultry, pigs, and ruminants."

The manuscript is well written and easy to comprehend. My main concern however, is that the study conclusion is baseless as there is no data, presented in the paper or previously published, to support it. Additionally, whilst it is appreciated that ST1193 is an important pathogenic E. coli ST, the authors have not articulated, why this study is important.

We hope that the significant edits made resolve these issues and provide clearer statements of importance in relation to the data presented.

1. L22: "Absent food animals" should be "absent in food animals"

The abstract was heavily re-written per other comments. This change was included.

2. L26-35: The importance of the study is unclear. The authors give a background on the distribution of F plasmids and a brief description of pUTI89. At L34-35 the authors state that "Our data suggests that senB+ plasmids are found in ExPEC lineages that colonise humans and companion animals but not E. coli that colonise food animals. " First, on reading the whole manuscript, this sentence is wrong as it is not backed by data. Seccon, even if it were the sentence were correct, there is nothing to show why this result and indeed the whole study are important, which what this section is about.

Importance has been re-written.

3. L28: Define RST on first time use

RST definition and naming convention has been provided.

4. L50: What evidence is there to suggest that ST1193 has never been recovered from poultry, pigs or ruminants? Did the authors review all ST1193 papers and found no reports of ST1193 from these animals? Even if no report of ST1193 in these animals was found, there has no evidence of sampling from such animals. None of the papers cited at L49 had ever attempted to sample poultry, pigs or ruminants. The lack of report of ST1193 from these animals may not necessary imply that these animals don't host ST1193, but rather, a result of sampling bias.

Additional exploration of data from EnteroBase (Supp. Table 2) and manuscript references have been included to support our reasoning and observations, and why we believe sampling bias is not a good reason for the lack of ST1193 data in animal agriculture.

5. L90:-93

a. "Most genes were detected using ABRicate...". What tool was used to detect "other (fewer)" genes?

More specific language has been used to avoid confusion.

b. Authors should also specify what genes were detected with ABRicate. Of course, from the mentioned databases [L91], what can infer that these were virulence, plasmid replicon and AMR genes. However, this may not be obvious to all readers and the authors should explicitly state the gene types they searched for.

Methods have been more clearly defined as suggested.

c. At L91, the authors mention ResFinder as one of the databases they used to search for genes using ABRicate. At L92-93, the author state that "ABRicate was run using default settings". The default database for ABRicate is the ncbi amr database. Can the authors clarify on which database they exactly used? If indeed the authors used ResFinder, what was their reason for not using the default (NCBI) database, which has more non-redundant AMR gene variant sequences than ResFinder?

That ResFinder was used has been made explicit. ResFinder was used particularly as our focus was mobilised elements in this paper, for which ResFinder is an appropriate database to use.

6. L93-94 and L203: If BLASTn was run without cut-off values, what was the threshold for determining presence or absence of class 1 integrase gene *int1*, *int1* truncations, and IS26? And how consistent is this with 208-209 where identity match of 70-100% are specified?

How additional BLASTn analysis was performed has been included. The 70-100% match mentioned is for a separate analysis generated for the BRIG (circular) visualisation and is separate to this analysis.

7. L116: "Phylogenetic distribution" should be "Phylogenetic tree"

Edit made.

8. L123: "Limited plasmid diversity" would be more appropriate than "lack of plasmid diversity" as there is some diversity.

Line removed to avoid confusion, with this point now covered by more direct observations of the data.

9. L226-L229 and L239: Include colour keys for Figure 4 and Figure 5 or describe what the colours represent in the respective figure legends.

Details added for colours within the figure descriptions.

10. L253-258: The absence of isolates from pigs cattle and other food animals is clearly due to sampling bias. The cited studies in this paper, which also contributed to the ST1193 genomes obtained from Enterobase for this study never attempted to sample from poultry, pigs or ruminants and thus as in my previous comment, the absence of genomes of isolates from these animals does not imply the animals don't carry *E. coli* ST1193.

Additional analysis and description has been included as stated for point 4.

11. L344-346: "Our data contribute to the hypothesis that *senB*-encoding F plasmids are found in ExPEC lineages that colonise humans or have close human association, such as companion animals and urbanised wildlife, but not *E. coli* that colonise poultry, pigs, and ruminants". This study has no data on poultry, pigs and ruminants hence has no basis to make the such a conclusion. L344-L346, should be deleted or amended to reflect the data reported in the manuscript.

As above, our words have been amended to reflect the data as suggested.

October 16, 2022

Prof. Steven Philip Djordjevic
Australian Institute for Microbiology and Infection
15 Broadway, University of Technology Sydney
Ultimo, NSW 2007
Australia

Re: Spectrum02554-22R1 (Global phylogeny and F virulence plasmid carriage in pandemic Escherichia coli ST1193)

Dear Prof. Steven Philip Djordjevic:

Editor's comments: Line 285 "ST1193 is a highly homologous pandemic lineage" homologous is not the correct term to describe lineages.

The last sentence in the Abstract needs to be revised because it does not reflect the conclusions of the study.

Thank you for submitting your manuscript to Microbiology Spectrum. As you will see your paper is very close to acceptance. Please modify the manuscript along the lines I have recommended. As these revisions are quite minor, I expect that you should be able to turn in the revised paper in less than 30 days, if not sooner. If your manuscript was reviewed, you will find the reviewers' comments below.

When submitting the revised version of your paper, please provide (1) point-by-point responses to the issues raised by the reviewers as file type "Response to Reviewers," not in your cover letter, and (2) a PDF file that indicates the changes from the original submission (by highlighting or underlining the changes) as file type "Marked Up Manuscript - For Review Only". Please use this link to submit your revised manuscript. Detailed instructions on submitting your revised paper are below.

Link Not Available

Sincerely,

Cheryl Andam

Preparing Revision Guidelines

- Point-by-point responses to the issues raised by the reviewers in a file named "Response to Reviewers," NOT IN YOUR COVER LETTER.
- Upload a compare copy of the manuscript (without figures) as a "Marked-Up Manuscript" file.
- Each figure must be uploaded as a separate file, and any multipanel figures must be assembled into one file.
- Manuscript: A .DOC version of the revised manuscript

- Figures: Editable, high-resolution, individual figure files are required at revision, TIFF or EPS files are preferred

Please return the manuscript within 60 days; if you cannot complete the modification within this time period, please contact me. If you do not wish to modify the manuscript and prefer to submit it to another journal, please notify me of your decision immediately so that the manuscript may be formally withdrawn from consideration by Microbiology Spectrum.

To the editor,

Thank you kindly for the last suggestions to our manuscript as it nears acceptance. The recommended changes are below as well as our edits made in response.

Yours,

Steven Djordjevic

1. Line 285 "ST1193 is a highly homologous pandemic lineage" homologous is not the correct term to describe lineages.

The line has been reworded to read "In summary, ST1193 is a pandemic lineage comprised of several serogroups, and while there was observable..."

2. The last sentence in the Abstract needs to be revised because it does not reflect the conclusions of the study.

Sentence has been reworded to the following: "Further epidemiological investigation of ST1193 should seek to confirm its presence in human-associated environments and identify any potential agricultural links, which are currently lacking."

November 3, 2022

Prof. Steven Philip Djordjevic
Australian Institute for Microbiology and Infection
15 Broadway, University of Technology Sydney
Ultimo, NSW 2007
Australia

Re: Spectrum02554-22R2 (Global phylogeny and F virulence plasmid carriage in pandemic Escherichia coli ST1193)

Dear Prof. Steven Philip Djordjevic:

Your manuscript has been accepted, and I am forwarding it to the ASM Journals Department for publication. You will be notified when your proofs are ready to be viewed.

Sincerely,

Cheryl Andam
Editor, Microbiology Spectrum

Journals Department
Supplemental Table 2: Accept
Supplemental Table 1: Accept
Supplemental Table Descriptions: Accept